# Anaerobic Speed Reserve, Sprint Force–Velocity Profile, Kinematic Characteristics, and Jump Ability among Elite Male Speed- and Endurance-Adapted Milers

**DOI:** 10.3390/ijerph19031447

**Published:** 2022-01-27

**Authors:** Pedro Jiménez-Reyes, Víctor Cuadrado-Peñafiel, Juan A. Párraga-Montilla, Natalia Romero-Franco, Arturo Casado

**Affiliations:** 1Centre for Sport Studies, Rey Juan Carlos University, E-28943 Madrid, Spain; peterjr49@hotmail.com (P.J.-R.); arturocasado1500@gmail.com (A.C.); 2Education Faculty, Autonomous University of Madrid, E-28049 Madrid, Spain; victoriefc@yahoo.es; 3Department of Didactics of Musical, Plastic and Corporal, University of Jaén, E-23071 Jaén, Spain; jparraga@ujaen.es; 4Nursing and Physiotherapy Department, University of the Balearic Islands, E-07122 Palma de Mallorca, Spain; 5Health Research Institute of the Balearic Islands (IdISBa), E-07010 Palma de Mallorca, Spain

**Keywords:** maximal force, performance, maximal power, middle-distance running

## Abstract

This study aimed to compare sprint, jump performance, and sprint mechanical variables between endurance-adapted milers (EAM, specialized in 1500–3000-m) and speed-adapted milers (SAM, specialized in 800–1500 m) and to examine the relationships between maximal sprint speed (MSS), anaerobic speed reserve (ASR), sprint, jump performance, and sprint mechanical characteristics of elite middle-distance runners. Fifteen participants (8 EAM; 7 SAM) were evaluated to obtain their maximal aerobic speed, sprint mechanical characteristics (force–velocity profile and kinematic variables), jump, and sprint performance. SAM displayed greater MSS, ASR, horizontal jump, sprint performance, and mechanical ability than EAM (*p* < 0.05). SAM also showed higher stiffness in the 40-m sprint (*p* = 0.026) and a higher ratio of horizontal-to-resultant force (RF) at 10 m (*p* = 0.003) and RFpeak (*p* = 0.024). MSS and ASR correlated with horizontal (*r* = 0.76) and vertical (*r* = 0.64) jumps, all sprint split times (*r* ≤ −0.85), stiffness (*r* = 0.86), and mechanical characteristics (*r* ≥ 0.56) during the 100-m sprint, and physical qualities during acceleration (*r* ≥ 0.66) and sprint mechanical effectiveness from the force–velocity profile (*r* ≥ 0.69). Season-best times in the 800 m were significantly correlated with MSS (*r* = −0.86). Sprint ability has a crucial relevance in middle-distance runners’ performance, especially for SAM.

## 1. Introduction

Performance in middle-distance runners is determined by tactical decision-making and physiological and mechanical factors [1]. Success in athletic races from 800 m to 3000 m is characterized by rapid, economical, and cyclical movement patterns [2]. Athletes need to sustain running velocities at and above maximal aerobic speed (MAS), deemed as the minimum speed at which maximum oxygen uptake is attained, and develop their sprinting ability to a great extent in order to achieve successful performances at major championships [3,4]. Although running economy and MAS are considered main middle-distance running performance determinants [5], recent studies also highlight the important role of anaerobic qualities [6,7] such as anaerobic speed reserve (ASR), which is the speed zone ranging from MAS to maximal sprint speed (MSS) [8,9]. Given that elite middle-distance runners display high levels of MAS [5] and anaerobic capacity, it seems that ASR should be considered to understand the underpinning mechanisms explaining their performance. In addition, MSS, which represents the upper part of the ASR spectrum, could also be an appropriate key performance parameter in middle-distance runners, not only because it allows athletes to achieve faster paces over longer distances [6] but also because it is the way to increase their ASR.

Furthermore, ASR has been proposed to be useful in categorizing different middle-distance runners’ profiles (i.e., 400–800 m, 800 m, and 800–1500 m types) and identifying training models (i.e., specific prescription of high-intensity training sessions) according to these profiles [10]. In this sense, different running categories exist in each distance event, and 1500 m runners can be considered either speed-adapted (800–1500 m specialists) or endurance-adapted (1500–3000 m specialists) milers (SAM and EAM, respectively) [11].

In addition, current evidence fails to describe the influence of anaerobic capacity (i.e., the ability to display higher speeds than MAS and lower than MSS) on running performance and ASR in middle-distance runners. In this term, the force–velocity (F-V) profile [12,13] during a maximum speed sprint would allow to describe the mechanical effectiveness in force application (i.e., the percentage of the resultant force that is produced in the horizontal direction) at the upper limit of ASR in middle-distance runners [14], apart from helping coaches to implement individualized training programs [12,13].

Apart from anaerobic factors, several studies have shown a rising interest in mechanical parameters such as the association of stiffness with running economy (RE) and maximal velocity [2,15], or the existing differences in running kinematics (i.e., step length, frequency, and flight and contact time) among athletes of different performance levels [16] and categories [11]. However, the influence of all these mechanical parameters on running performance in middle-distance running events has not been explored sufficiently yet. Since increasing evidence suggests that performance in these events could be strongly connected to anaerobic characteristics and sprint ability [7,17], it would be useful for athletes and coaches to describe the aforementioned mechanical parameters in middle-distance runners and elucidate the relationship to performance determinants.

Therefore, the aims of the present study were: (1) to describe and compare sprint and jump performance and sprint mechanical variables between elite male SAM and EAM, and (2) to examine the relationships between MSS and ASR with sprint and jump performance and sprint mechanical variables in elite male middle-distance runners. We hypothesize that SAM will display greater sprint and jump performance and more efficient sprint mechanical responses than EAM and that MSS and ASR will significantly correlate with sprint and jump performance and efficiency of sprint mechanical responses.

## 2. Materials and Methods

### 2.1. Participants

A convenient sample of 15 elite male middle-distance running athletes (age = 24.5 ± 4.6 years, body mass = 63.1 ± 4.3 kg, height = 1.77 ± 0.05 m) voluntarily participated in this study. All athletes trained in the same training group and shared the same coach at the Sport High Performance Centre of the Spanish Government (Madrid). To conduct an exhaustive analysis, participants were classified based on their coach’s perspective regarding the athletic event they were specialized in. Accordingly, eight and seven runners were considered SAM and EAM, respectively. This coach’s decision was based on the target event for the season and the training characteristics being used (i.e., milers who were also targeting the 800 m event displayed higher intensity and lower volume in training; milers who were also targeting the 3000 m, 3000 m steeplechase, or 5000 m events displayed lower intensity and higher volume in training). Additionally, this decision was confirmed by means of analyzing the difference in recent competitive performances between shorter (i.e., 800 m) and longer (i.e., 3000 m) events than 1500 m. The World Athletic open access website was used to collect the best performances achieved by participants during competition 12 months prior to testing (www.worldathletics.org (accessed on 15 December 2021)). These results were transformed into World Athletics (WA) scores [18]. If the “better” recent performance of participants was achieved in shorter events than 1500 m, they were allocated to the SAM group, and if the “better” recent performance was shown in longer events than 1500 m, they were allocated to the EAM group.

Eligibility criteria required that participants: (a) were current elite athletes, considered those who competed internationally or at senior category national championships, to accord with widely used criteria used to define elite athletes [19]; (b) had at least 5 years of systematic training experience (Table 1); (c) were free from health problems or musculoskeletal injuries that could compromise testing performance during at least 6 months prior to the beginning of the study. All the participants were informed of the study procedures and signed a written informed consent form prior to initiating the study. The study protocol adhered to the tenets of the Declaration of Helsinki and was approved by the Institutional Review Board of Pablo de Olavide University (935/CEIH/2019).

### 2.2. Study Design

A cross-sectional study was designed in two different testing sessions, with 1 week apart: (1) participants performed an incremental treadmill test through which physiological performance outcomes were obtained; and (2) participants performed a testing battery in this order: maximal vertical and horizontal jumps and maximal 40 m and 100 m sprints. Prior to the beginning of the second testing session, all athletes performed a 15-min warm-up and a familiarization process to ensure the correct execution of all testing procedures. In addition, athletes were asked to avoid intense exercise in the 24 h before testing, apart from ensuring that the training load was similar for all athletes in the last 3 days prior to testing to avoid fatigue-affected results. These sessions were carried out in the High Performance Centre of the city in June 2019, which refers to the competitive period of all the athletes participating in the study.

### 2.3. Testing Session 1

Maximal aerobic speed (MAS): The maximal aerobic speed was determined through an incremental treadmill test (Technogym, Exite Run 600, Cesena, Italy) that was performed 1 week before the aforementioned testing session. All participants carried out a standardized warm-up consisting of low-intensity running and, thus, started the test at 8.0 km·h^−1^, with 2% as the treadmill slope and progressive increments of 0.5 km·h^−1^ every 30 s until exhaustion. Exhaustion was considered when the runners volitionally declared their incapacity to continue at the predetermined pace. During the test, gas analyses were conducted since the participants breathed through a low dead space (90 mL), low resistance (5.5 cm H_2_O at 510 L·min^−1^) facemask, and turbine assembly. Gases were drawn continuously from the facemask to a breath-by-breath gas analyzer (Fitmate Pro, Cosmed, Rome, Italy) through a 2 m sampling line (0.5 mm internal diameter) and were analyzed for O_2_ and CO_2_ (with a 200 ms delay). A turbine volume transducer (Interface Associates, Alifovieja, CA, USA) determined the expired volumes. Prior to each test, the breath-by-breath gas analyzer was calibrated by using gas mixtures (Linde Gas, London, UK) of concentrations previously known. The turbine was also calibrated prior to each test with a 3 L calibration syringe (Hans Rudolf, Shawnee, KS, USA). Oxygen uptake was calculated and displayed on a breath-by-breath basis. A computer was used to integrate the volume and concentration signals by converting values from analog to digital format. In this conversion, the delay of the gas transit through the capillary and the room temperature was taken into account. MAS was considered as the slowest speed at which maximum oxygen uptake was attained [5].

Anaerobic speed reserve (ASR): ASR was calculated as the difference between MAS and MSS. ASR is a good reflection of the speed range that an athlete possesses, considering these two milestones. With this information, it is possible to calculate the speed reserve ratio (SRR) as the coefficient of “maximal sprint speed (km·h^−1^)/maximal aerobic speed (km·h^−1^)”.

### 2.4. Testing Session 2

All participants performed a 15-min warm-up consisting of 5 min of jogging and 5 min of lower limb dynamic stretching. During the last 5 min, as part of the specific warm-up, participants also performed three progressive sprints of 40 m at 50%, 70%, and 90% effort. As a familiarization process, all athletes performed progressive trials in the case of jumps and progressive accelerations from the starting line in the case of races.

Vertical jump—countermovement (CMJ): Just after warming up and familiarization, the runners carried out a maximum vertical jump. They started from an upright position, with their hands on their waists. Thus, they performed a countermovement by flexing their knees up to 90° and jumping as high as possible. During the flight phase of the jump, they should maintain their knees extended up to 180°, without hyperextending their hips [20].

Horizontal jump—standing long jump (SLJ): Athletes were instructed to perform a maximal horizontal jump from a starting line, with both feet simultaneously and arms swinging, and without a run-up. The maximal metered performance was measured by taking into account the rear part of the most indented heel [21].

For both vertical and horizontal jumps, participants performed three trials, with 2 min as the inter-trial passive recovery, and the best one was recorded (in meters). A sports professional external to the investigation supervised the correct execution of jumps, and an OptoGait Photoelectric Cell System (OptoGait, Microgate, Bolzano, Italy) was used.

40 m sprint: After 4 min of rest, participants performed three maximal sprints of 40 m, with 4 min as the inter-trial period of rest. The fastest one was considered for the analysis. Athletes were instructed to start from a crouching position (staggered stance). A previously validated simple field method was used to compute sprint performance and mechanical outputs [13]. A Stalker Acceleration Testing System (ATS) II radar device (Stalker ATS II, Applied Concepts, Dallas, TX, USA) at 46.9 Hz was employed to collect velocity–time data of each sprint. The radar device was attached to a tripod 10 m from the starting line at a height of 1 m, which corresponded to the height of participants’ center of mass. Based on Samozino’s method, sprint mechanical variables were obtained from the velocity–time data [13,14]. This validated method is a macroscopic biomechanical model to estimate external horizontal force production during sprinting from the velocity of the center of mass using the inverse dynamic approach [14]. The outcomes regarding the F-V profile were: maximal theoretical force (F0), maximal theoretical velocity (V0), F-V slope, maximal power (Pmax), decrease in the ratio of horizontal-to-resultant force (DRF), maximal ratio of horizontal-to-resultant force (RFpeak), and this same variable at 10 m (RF_10m). DRF and RFpeak are commonly employed to assess mechanical effectiveness and have been correlated with sprint performance. From the data of this test, we also calculated MSS. We also obtained sprint split times at 10 m, 20 m, 30 m, and 40 m.

100 m sprint: After 4 min of rest, athletes performed two maximal sprints of 100 m from a crouching position (staggered stance), with 10 min of rest between sprints. For stride pattern/mechanical variables and partial times, an optoelectronic system (Optojump Next Microgate, Bolzano, Italy) was installed on the lane of an indoor track to obtain running kinematics from 30 to 40 m during the maximum velocity phase and from 80 to 90 m during the decrement of velocity phase. This material permits measurement of contact time on the floor (CT), flight time (FT), step time (ST), stride length (SL), stride flight (SF), and step velocity (SV). SL asymmetry (SLasy) was calculated as the absolute difference of distance covered on three left-foot strides minus the distance covered on three right-foot strides. SR, SL, FT, and CT were measured and averaged from the third up to the eighth last stride of the approach. Furthermore, partial times were collected for 21-to-30 m, 30 m, 60 m, 80 m, and 100 m. For vertical and leg stiffness, step characteristics of CT and FT sampled at 1000 Hz were captured via a series of ground-based photoelectric cells (Microgate: OptoJump, Bolzano, Italy) positioned between 40 and 49 m of the sprint track. This part of the sprint is often the segment where runners achieved Vmax (as shown in pilot testing). The spring–mass model [22,23] was used to compute the mechanical leg behavior during the ground contact. The calculation of peak vertical ground reaction force (Fmax), vertical stiffness (Kvert), and leg stiffness (Kleg) was based on the method validated by Morin et al. (2005) [24].

### 2.5. Statistical Analyses

Descriptive data are presented as means and standard deviations. The degree of the linear relationship between variables was examined using Pearson’s product moment correlation. Independent sample *t*-tests and Cohen’s *d* effect size (ES) with 95% confidence intervals were used to compare the sprint mechanical F-V profile (F0, V0, Pmax, DRF, and RFpeak), ASR, mechanical variables of stride patterns, and MSS between types of middle-distance runners. The scale used for interpreting the magnitude of the effect size was specific to training research: negligible (<0.2), small (0.2–0.49), moderate (0.5–0.79), and large (≥0.8) [25]. Statistical significance was set at *p* ≤ 0.05. Data were analyzed using SPSS 20.0 software (SPSS Inc. Chicago, IL, USA) and Office Excel 2010 (Microsoft Corporation, Redmond, WA, USA).

## 3. Results

Table 2 shows the descriptive outcomes of aerobic–anaerobic variables, sprint mechanical profile, and performance of SAM and EAM. Significant differences were found between groups in variables describing aerobic–anaerobic performance, SLJ distance, 100 m sprint performance, and sprint mechanical profile and effectiveness. In all variables, SAM displayed a greater performance than EAM.

Table 3 shows the correlations between MSS and ASR, and the remaining sprint mechanical characteristics and physiological performance determinants derived from the incremental treadmill test that participants carried out in the study. Season-best performances in 800 m showed a significantly negative and small-to-moderate correlation with ASR (*p* < 0.05). ASR and MSS showed positive and moderate-to-high correlations with performance in both maximal jumps, mainly for results related to SLJ (*r* > 0.75, *p* < 0.01) (Table 3). It is also important to highlight the significantly high-to-very high negative correlation (*r* > −0.7, *p* < 0.01) observed between ASR and MSS and all 100 m sprint split times, especially those between MSS and split times belonging to longer distances than 30 m (*r* > −0.9, *p* < 0.001) (Table 3). ASR and MSS showed significantly positive and moderate-to-very high correlations with all F-V profile variables (*r* > 0.6, *p* < 0.05), except for F-V profile and DRF, which did not show any significant correlation (*p* > 0.05). In addition, the correlations observed between both ASR and MSS and V0 were very high and positive (*r* > 0.9, *p* < 0.001) (Table 3). No other significant correlations were found (*p* > 0.05).

## 4. Discussion

The main finding of this study was a large significant correlation between 800 m performance and MSS, despite the lack of significant correlations between 800 m or 1500 m performance with ASR. In addition, jumping ability, stiffness, 100 m sprint performance, and F-V mechanical determinants were highly correlated with both ASR and MSS.

The large correlation observed between MSS and 800 m performance is in line with findings from Sandford et al., who reported an influence of ASR on the variability of running performance in elite 800 m runners when assuming similar MAS values and, therefore, the ASR was determined by MSS [7]. Although results from the present study indicate that ASR was not significantly correlated with 800 m performance, the correlation was higher than that observed with 1500 m performance. If we consider that the use of aerobic metabolism is greater with event distance (66% in 800 m vs. 88% in 1500 m) [26], the influence of ASR on performance was lower in 1500 m than 800 m in the present study. Although these results would indicate that measuring ASR might be useless [27], ASR has been found effective in prescribing individualized training programs at higher intensities than MAS [28]. Additionally, MSS was correlated with 800 m performance, representing a performance determinant, as previous studies showed in longer distance running events such as 5000 m [29] and 10,000 m [17]. Although our results did not report a significant correlation between MSS and 1500 m performance, it has been previously noted that pacing profiles and tactical behaviors in middle-distance running races can be very different during “meet” than major championship races. In the case of “meet” races, the goal is to achieve the overall fastest possible performance, while during major championships, athletes aim to achieve the highest finishing position [30]. In this sense, championship races are characterized by slower races than “meets,” with a fast endspurt [3,4]. Therefore, developing MSS during training is also important in elite middle-distance runners, especially for 800 m.

In addition, SAM showed longer SLJ than EAM, and SLJ and CMJ performances significantly correlated with both MSS and ASR. These findings match with those from Maulder and Cronin, who found that both vertical and horizontal jump abilities correlated with sprint performance in the male sports population [21].

In the case of the 100 m sprint, although no differences were observed between SAM and EAM in any of the kinematic variables, ST and SF showed significant correlations with MSS and ASR and SV and CT with MSS. In line with our results, Brughelli et al. reported that rugby players increased their stride rate and SF and reduced their CT as long as they were reducing their speed when subsequently conducting sprinting bouts of increasing distances [31]. The relationships found between kinematic variables and ASR in the present study may be associated with the greater ASR found in SAM compared with EAM. In this line, 100 m performance was greater in SAM and highly correlated with both MSS and ASR. These findings could be explained by a greater MSS observed in SAM compared with EAM, while MAS remained similar for both groups. The differences found in MSS between both groups agree with the findings by Casado et al. [11].

In the same line, SAM displayed a greater stiffness than EAM. This finding would be expected considering the higher MSS displayed by 800 m runners than that in 1500 m runners [6,7,32]. Our results also showed that Kvert significantly correlated with both MSS and ASR. These findings match with previous research that found similar values in various running conditions [15,33,34]. Our results also agree with those from previous studies that confirmed the important role of stiffness in running economy [15,35,36], thereby reducing the energy cost for a given velocity and MSS.

If we consider the F-V profile components, SAM displayed higher values for F0, V0, Pmax, Power Pmax, RF_10m, and RFpeak than EAM, with large significant correlations between these variables and both MSS and ASR. In this line, previous studies have shown that a higher F0 was more effective on the development of short accelerations in hurdlers and sprinters [37,38]. As we aforementioned, middle-distance runners with high F0 may be able to perform sudden accelerations during championship races to successfully cope with tactical strategies during championship races [3,30]. Similarly, V0 has been associated with the ability to produce long accelerations and reach a high sprint velocity [12], which would be paramount during middle-distance running races [3,4]. In addition, DRF and RFpeak characterize mechanical effectiveness during sprint running [14].

The main limitation of the present study refers to the small sample size. However, the sample quality is high, given the outstanding performance level of participants. A secondary limitation of the present study relies on its observational character rather than interventional. In this sense, further studies should focus on implementing different training characteristics that may influence the development of sprint performance and that of its mechanical determinants, leading to performance improvement in middle-distance runners.

As practical applications, coaches should consider monitoring and developing sprinting ability in middle-distance runners to improve 800 m performance during championship races. To this end, it is important to decrease contact time and increase stiffness and stride flight time during sprint by monitoring 100 m performance, jumping ability, and F-V profiles across the season. The development of these parameters may be associated with an improved ability to generate a higher speed during critical moments of a race, with greater mechanical effectiveness at high velocities.

## 5. Conclusions

Elite male SAM displayed better horizontal jump and sprint performances than elite male EAM, as well as greater efficiency in sprint mechanical variables such as stiffness, physical qualities during acceleration, and mechanical effectiveness. These differences were not shown in vertical jump performance and stride pattern during sprint. In addition, MSS and ASR highly correlated with sprint and jump performances, stiffness, physical qualities during acceleration, and those related to mechanical effectiveness. Some kinetic variables during sprint such as ST and SF also correlated with MSS and ASR, and SV and CT only did so with MSS. A large correlation was shown between 800 m performance and MSS.

Practitioners should consider that sprinting ability is strongly related to performance in middle-distance runners, especially in those specialized in shorter races like the 800 and 1500 m.

## Figures and Tables

**Table 1 ijerph-19-01447-t001:** Performance characteristics of participants.

Competitive Level	Greatest Competition in Which Athletes Participated	Other Relevant Information	World Athletics Scores	SB (min:s.cs)
			Mean ± SD	Mean ± SD
**World (n = 2)** **European (n = 11)** **National (n = 2)**	**Olympic Games** **European Championships** **National Championships**	2 x national record holders 3 European medalists 2 x World Championship finalists 7 national champions	800 m (n = 7): 987.4 ± 120.1 1500 m (n = 15): 1011.6 ± 86.0 3000 m (n = 5): 974.0 ± 168.2	800 m (n = 7): 1:51.24 ± 4.29 1500 m (n = 15): 3:47.44 ± 6.8 3000 m (n = 5): 8:32.45 ± 27.5

SB, season-best time; SD, standard deviation.

**Table 2 ijerph-19-01447-t002:** Aerobic–anaerobic variables, sprint and jump performance, and sprint mechanical characteristics of participants in the study.

			SAM Athletes (n = 7)			EAM Athletes (n = 8)		
			Mean	±	SD	Mean	±	SD
**Aerobic–anaerobic**		**MSS (km/h)**	33.02	±	1.19	30.66 **^d^	±	1.18
		**vVO_2max_**	21.90	±	0.59	21.96	±	0.98
		**ASR (km/h)**	11.12	±	0.98	8.71 **^d^	±	1.48
		**SRR**	1.51	±	0.05	1.40 **^b^	±	0.08
**Maximal jump**		**CMJ (cm)**	39.64	±	6.51	34.49	±	6.31
		**SLJ (m)**	2.40	±	0.19	2.16 *^b^	±	0.14
**100 m sprint**	**Stride pattern (mechanical variables)**	**CT (s)**	0.110	±	0.008	0.118	±	0.009
		**FT (s)**	0.133	±	0.011	0.133	±	0.008
		**ST (s)**	0.243	±	0.013	0.251	±	0.008
		**SL (m)**	2.103	±	0.125	2.083	±	0.126
		**SF (m)**	4.126	±	0.221	3.983	±	0.134
		**SV (m/s)**	8.663	±	0.573	8.260	±	0.452
	**Time 100 m**	**21_30 m (s)**	0.98	±	0.04	1.03 *^a^	±	0.04
		**30 m (s)**	4.22	±	0.12	4.50 **^b^	±	0.15
		**60 m (s)**	7.44	±	0.24	7.84 *^b^	±	0.27
		**80 m (s)**	9.62	±	0.34	10.20 **^c^	±	0.33
		**100 m (s)**	11.84	±	0.45	12.61 **^c^	±	0.43
	**Stiffness**	**Fmax (BW)**	3.48	±	0.18	3.25	±	0.30
		**Kvert (kN/m)**	79.58	±	11.21	67.57 *^d^	±	7.20
		**Kleg (kN/m)**	14.20	±	1.65	12.32	±	2.14
**40 m sprint (F-V profile)**	**Physical qualities during acceleration**	**V0 (m/s)**	9.68	±	0.34	8.95 **^c^	±	0.34
		**F0 (N/kg)**	7.18	±	0.34	6.58 *^c^	±	0.57
		**Pmax (W/kg)**	17.24	±	1.30	14.62 **^d^	±	1.57
		**FV-profile**	−47.10	±	4.38	−46.13	±	3.67
	**Mechanical effectiveness**	**RF_10m**	0.336	±	0.015	0.304 **^a^	±	0.018
		**RF_peak**	0.500	±	0.020	0.466 *^a^	±	0.029
		**DRF**	−0.066	±	0.003	−0.066	±	0.005

ASR, anaerobic speed reserve; CMJ, countermovement jump; CT, contact time; DRF, decrease in the ratio of horizontal-to-resultant force; EAM, endurance-adapted athletes (1500–3000 m); F0, maximal theoretical force; Fmax, peak vertical group reaction force; FT, flight time; FV, force–velocity; Kleg, leg stiffness; Kvert, vertical stiffness; MSS, maximal sprint speed; Pmax, maximal power; RF, ratio of horizontal-to-resultant force; SAM, sprint-adapted athletes (800–1500 m); SD, standard deviation; SLJ, single long jump; SRR, sprint reserve ratio; ST, step time; SL, stride length; SF, stride flight; SV, step velocity; V0, maximal theoretical velocity; vVO_2max_, velocity at maximal oxygen uptake. Scheme 0. * *p* < 0.05; ** *p* < 0.01; ^a^ negligible effect size (<0.2); ^b^ small effect size (0.2–0.49); ^c^ moderate effect size (0.5–0.79); ^d^ large effect size (≥0.8).

**Table 3 ijerph-19-01447-t003:** Correlations between Maximal Sprint Speed and Anaerobic Speed Reserve with sprint, jump performance and sprint mechanical characteristics of participants.

			MSS		ASR	
			r	(95%CI)	r	(95%CI)
**Running performance (SB)**		**800 m**	–0.86 *	(–0.23, –1.0)	–0.62	(–0.66, 1.0)
		**1500 m**	–0.05	(–0.58, 0.53)	0.33	(–0.26, 0.76)
**Maximal jump**		**CMJ (cm)**	0.64 *	(0.23, 0.88)	0.70 *	(0.36, 0.90)
		**SLJ (m)**	0.76 **	(0.49, 0.94)	0.80 **	(0.60, 0.93)
**100 m sprint**	**Step (maximal sprint speed)**	**CT (s)**	–0.58 *	(0.04, 0.81)	–0.42	(–0.77, 0.13)
		**FT (s)**	–0.34	(–0.82, 0.39)	–0.35	(–0.78, 0.35)
		**ST (s)**	–0.72 **	(–0.31, –0.94)	–0.61 *	(–0.12, –0.92)
		**SL (m)**	0.17	(–0.42, 0.73)	0.0	(–0.55, 0.61)
		**SF (m)**	0.73 **	(0.32, 0.94)	0.61 *	(0.13, 0.93)
		**SV (m/s)**	0.70 **	(0.29, 0.91)	0.48	(0.08, 0.82)
	**Time 100 m**	**21_30 m (s)**	–0.85 ***	(–0.59, –0.96)	–0.70 **	(–0.42, –0.88)
		**30 m (s)**	–0.93 ***	(–0.84, –0.98)	–0.93 ***	(–0.82, –0.98)
		**60 m (s)**	–0.91 ***	(–0.78, –0.97)	–0.88 ***	(–0.72, –0.97)
		**80 m (s)**	–0.96 ***	(–0.91, –0.99)	–0.90 ***	(–0.76, –0.95)
		**100 m (s)**	–0.94 ***	(–0.87, –0.99)	–0.82 ***	(–0.69, –0.93)
	**Stiffness**	**Fmax (BW)**	0.42	(–0.20, 0.77)	0.31	(–0.33, 0.72)
		**Kvert (kN/m)**	0.86 ***	(0.65, 0.96)	0.72 **	(0.49, 0.88)
		**Kleg (kN/m)**	0.51	(0.10, 0.76)	0.47	(–0.04, 0.77)
**40 m sprint (F-V profile)**	**Physical qualities during acceleration**	**V0 (m/s)**	1.00 ***	(0.99, 1.0)	0.91 ***	(0.84, 0.97)
		**F0 (N/kg)**	0.66 *	(0.34, 0.94)	0.75 **	(0.42, 0.95)
		**Pmax (W/kg)**	0.88 ***	(0.74, 0.97)	0.90 ***	(0.75, 0.98)
		**FV-profile**	–0.22	(–0.79, 0.37)	–0.34	(–0.81, 0.21)
	**Mechanical effectiveness**	**RF_10m**	0.89 ***	(0.74, 0.97)	0.91 ***	(0.75, 0.99)
		**RF_peak**	0.69 **	(0.34, 0.92)	0.77 **	(0.45, 0.96)
		**DRF**	0.19	(–0.47, 0.62)	–0.23	(–0.58, 0.45)

ASR, anaerobic speed reserve; CMJ, countermovement jump; CT, contact time; DRF, decrease in the ratio of horizontal-to-resultant force; EAM, endurance-adapted athletes (1500–3000 m); F0, maximal theoretical force; FMax, peak vertical group reaction force; FT, flight time; FV, force–velocity; Kleg, leg stiffness; Kvert, vertical stiffness; MSS, maximal sprint speed; Pmax, maximal power; RF, ratio of horizontal-to-resultant force; SAM, sprint-adapted athletes (800–1500 m); SB, season best; SLJ, single long jump; ST, step time; SL, stride length; SF, stride flight; SV, step velocity; V0, maximal theoretical velocity; * *p* < 0.05; ** *p* < 0.01; *** *p* < 0.001.

## Data Availability

The data presented in this study are available on request from the corresponding author. The data are not publicly available due to privacy.

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
