# Peer review of "Anaerobic Speed Reserve, Sprint Force–Velocity Profile, Kinematic Characteristics, and Jump Ability among Elite Male Speed- and Endurance-Adapted Milers"

_ijerph, 2022, doi:10.3390/ijerph19031447_

Round 1

Reviewer 1 Report

Some minor mistakes should be corrected and it will be ready to publish:

Line 42: "...qualities [6,7]. such ...!

Line 141: same format for L.min-1.

Line 156:  same format for (km/h)

Line 157:  same format for (km/h)

Table 2: Notes: the significance of vVO2max

Reviewer 2 Report

The main aim of the paper „Anaerobic speed reserve, sprint force-velocity profile, kinematic characteristics and and jump ability among elite male speed- and endurance-adapted milers“ is to compare sprint, jump performance and sprint mechanical variables between the middle- and long-distance runners, and to examine the relationships between maximal sprint speed, anaerobic speed reserve, sprint, jump performance and sprint mechanical characteristics of elite distance runners.

I would like to acknowledge the efforts of the authors, the article is interesting. However, there are some points that need to be improved.

Major issues:

How were the participants chosen? Were they all members of one training group? Did they all have the same coach?

Was the training load content of the EAM and SAM group monitored? If so, were there significant differences in the volume of speed and endurance preparation?

Was the training program monitored last 3 days prior to testing? Did they all have the similar training load?

I recommend extending the conclusion, at least by commenting on the stated aims of the work. In this way, it is not indicative of the content and objective of the article.

I recommend extending the conclusion, at least by commenting on the stated aims of the work. In this way, it is not reflect of the content and the aim of the article.

Minor issues:

Line 135: missing superscript

Line 141: H2O - missing subscript

Line 144: CO2 and H2O - missing subscript

Reviewer 3 Report

Thank you for the opportunity to review this manuscript. The topic of the paper is interesting and fits the scope of the journal. The text is relatively well written and composed.

Major comments

The first aim of this study was to describe and compare sprint and jump performance and mechanical variables between elite male SAM and EAM. Why do you believe that maybe EAM can have better sprint and jump performance than SAM? in SAM group belong the speed-adapted runners that means they trained in higher intensity and they are faster than EAM. Thus, they will have better sprint and jump performance than EAM.

Lines 74-77. Please write better the aim of the study 

Lines 164-168. Please refer the reference that used for the CMJ.

Lines 169-172. Please refer the reference that used for the SLJ.

Minor comments

Line 3. Please remove the repeated “and” from title.

Line 42. Please remove the dot after “[6,7].

Line 45. Please remove the “6” from the capacity.

Reviewer 4 Report

First, recognize the merit of the work carried out by the authors. In my view, the research problem should be clearer in the introduction. The theoretical foundation is adequately done. The objectives are defined, however I question whether it was possible to establish study hypotheses.

The methodology is explained clearly and objectively.

In the results presentation, the tables appear outside the limits and the notes of the tables should be revised in order to be more perceptible (eg. a, b, c, d relative to effect sizes).

In the conclusions I think that they should be further developed, taking into account the objectives and results obtained.

Round 2

Reviewer 2 Report

I appreciate the efforts of the authors in writing the data processing and writing the interesting paper „Anaerobic speed reserve, sprint force-velocity profile, kinematic characteristics and and jump ability among elite male speed- and endurance-adapted milers“.

The submitted comments were incorporated into the text, I have no further comments on the evaluated contribution.

Reviewer 3 Report

The paper can be accept in present form.